# Design, Synthesis and Biological Evaluation of Novel Phenyl-Substituted Naphthoic Acid Ethyl Ester Derivatives as Strigolactone Receptor Inhibitor

**DOI:** 10.3390/ijms25073902

**Published:** 2024-03-31

**Authors:** Lin Du, Xingjia Li, Yimin Ding, Dengke Ma, Chunxin Yu, Hanqing Zhao, Ye Wang, Ziyan Liu, Liusheng Duan

**Affiliations:** 1State Key Laboratory of Plant Physiology and Biochemistry, Engineering Research Center of Plant Growth Regulator, Ministry of Education & College of Agronomy and Biotechnology, China Agricultural University, Beijing 100193, China; s20183101661@cau.edu.cn (L.D.); ymyf12345678@163.com (Y.D.); mdk@cau.edu.cn (D.M.); 2College of Plant Science and Technology, Beijing University of Agriculture, Beijing 102206, China; li1257391684@163.com (X.L.); cxyu0303@foxmail.com (C.Y.); zhaohanqingbua@163.com (H.Z.); wangyebua@126.com (Y.W.); liuziyan8028@163.com (Z.L.)

**Keywords:** strigolactone inhibitor, tillering, seed germination, senescence, molecular design

## Abstract

Strigolactones (SLs) are plant hormones that regulate several key agronomic traits, including shoot branching, leaf senescence, and stress tolerance. The artificial regulation of SL biosynthesis and signaling has been considered as a potent strategy in regulating plant architecture and combatting the infection of parasitic weeds to help improve crop yield. DL1b is a previously reported SL receptor inhibitor molecule that significantly promotes shoot branching. Here, we synthesized 18 novel compounds based on the structure of DL1b. We performed rice tillering activity assay and selected a novel small molecule, **C6**, as a candidate SL receptor inhibitor. In vitro bioassays demonstrated that **C6** possesses various regulatory functions as an SL inhibitor, including inhibiting germination of the root parasitic seeds *Phelipanche aegyptiaca*, delaying leaf senescence and promoting hypocotyl elongation of *Arabidopsis*. ITC analysis and molecular docking experiments further confirmed that **C6** can interact with SL receptor proteins, thereby interfering with the binding of SL to its receptor. Therefore, **C6** is considered a novel SL receptor inhibitor with potential applications in plant architecture control and prevention of root parasitic weed infestation.

## 1. Introduction

SLs are carotenoid derivatives that are widely recognized as endogenous plant hormones and rhizospheric signaling molecules; they play a key role in regulating shoot branching/tillering, root structure and other aspects of plant growth, in addition to contributing to pathogen defense and responses to abiotic stress [1,2,3,4]. SLs are primarily synthesized in the roots of plants, but there is also a small amount of SLs in stems and leaves. Root-secreted SLs serve as signaling molecules that aid plants in acquiring sufficient nutrients by stimulating branching of arbuscular mycorrhizal fungi (AMF) [5,6]. Simultaneously, released SLs also function as germination stimulants, inducing the germination of root parasitic weed seeds such as *Striga* and *Orobanche* [7]. Once germinated, these insidious weeds voraciously siphon off essential nutrients from their unsuspecting host plants, resulting in heavy burden on yield losses in multiple crop species [8,9]. It has been reported that *Striga* infestation caused nearly $700 million in annual economic losses and affected the livelihood of 300 million people [9]. Since all these biological effects of SLs are closely associated with crop yield, precise control of SL signaling could be an innovative tool for the improvement of crop production [10,11].

Chemical manipulation of plant hormones exhibits significant potential in agriculture, whether through the discovery of functional analogs or SL inhibitors [10,12,13]. Several SL analogs, such as GR24, SPL7, and 2NOD, have been identified to date for their ability to inhibit branching and stimulate the germination of parasitic weed seeds (Figure 1) [14,15,16]. Despite the promising properties of SL analogs, their widespread application in practical agricultural settings is hindered by factors such as application timing and drug stability [17]. As a result, various small molecules, including SOP, DL1b, and KK094, have been developed as SL antagonists (Figure 1) [13,18,19]. These antagonists often feature straightforward chemical structures that are easily synthesized and exhibit unique chemical frameworks. Consequently, SL inhibitors hold potential as regulatory agents for enhancing plant architecture and suppressing the germination of root-parasitic weed seeds, ultimately leading to increased crop yield.

The recognition of SLs is mediated by a substantially conserved group of α/β-fold hydrolase receptors [20,21]. However, there exist species-specific variations in SL receptors, exemplified by DAD2 in petunia, OsD14 in rice, AtD14 in *Arabidopsis*, and ShHTLs in *Striga* [22,23,24]. Through the utilization of molecular docking and structure–activity relationship (SAR) analysis experiments, DL1b has been previously identified as an inhibitor of SL receptor D14, exhibiting high branching activity [19,25]. Furthermore, the SAR analysis of DL1b demonstrated that the substitution of bromonaphthalene played a key role in the significant enhancement of D14 inhibitory activity [19]. Furthermore, the benzene ring is regarded as a crucial active scaffold in the drug development process [26,27]. For example, aspirin, paracetamol, ibuprofen, and imatinib (Figure 2) were all developed by introducing a benzene ring into the structure backbone; this improves the properties of drugs or enhances the drug’s target specificity [28,29,30,31]. These examples demonstrate the significance of introducing benzene ring structural fragments in drug discovery, as it often leads to improved pharmacological activity, target specificity, and overall drug efficacy. As a result, utilizing DL1b as a lead compound, we have successfully synthesized a series of novel candidates for SL inhibitors through the incorporation of benzene ring structures with diverse substituents, while concurrently preserving the original fragment of bromonaphthalene structure (Figure 2). The biological activities of these compounds were investigated by testing activities in shoot branching, parasitic weed seed germination, leaf senescence, and hypocotyl elongation. And the mechanism of action was determined based on the results of ITC and molecular docking analysis.

## 2. Results and Discussion

### 2.1. Chemistry

The synthetic route of target compounds **C1**–**C18** is shown in Figure 1 [25,32]. Starting from acetophenones and Br_2_, partial 2-Bromo-1-phenylethanones intermediates were successfully prepared by an electrophilic substitution. Subsequently, nucleophilic substitution of 2-Bromo-1-phenylethanones and 5-Bromo-1-naphthoic acid led to target compound **C1**–**C18**. Notably, all target compounds were successfully synthesized with an acceptable yield.

### 2.2. Biological Analysis

#### 2.2.1. Effect of the Target Compounds on Rice Tillering of Wild Type and *d10* Mutant

As previously reported, carotenoid cleavage dioxygenase CCD8 plays a crucial regulatory role in the biosynthetic pathway of SLs in plants [33,34]. To test their capability in regulating shoot branching, we applied the 18 target compounds at a concentration of 10 μM to hydroponically grown seedlings of the rice wild type (*Nipponbare*) and its mutants *d10*; an SL deficient, high-tillering dwarf mutant disrupted in the CCD8 gene. Here, DL1b was used as a negative control and DMSO was used as a positive control. As depicted in Figure 3A, not all compounds can promote tillering in rice plants. Among them, **C6** exhibited superior tillering-promoting activity, with a tiller number of 3.2 at 10 μM; significantly higher than the blank control and DL1b. Therefore, we further evaluated the effects of **C6** at 1 and 10 μM on tillering in wild-type rice and *d10* mutants to preliminarily assess its potential application as an SL receptor antagonist, as shown in Figure 3B,C. GR24, an analogue of SL, showed a statistically significant effect in inhibiting tillering of the SL deficient *d10* mutant, decreasing the number of its tillers from 2.4 to 1. However, the application of **C6** can effectively alleviate the inhibitory effect of GR24 on tillering in the *d10* mutant, increasing the tiller number from 1 to 2.1 and 2.3 at concentrations of 1 uM and 10 uM, respectively, which is consistent with the *d10* mutant. Meanwhile, we evaluated the effect of **C6** on dry biomass accumulation in wild-type rice and found that both **C6** and DL1b significantly increased the accumulation of dry biomass in rice. These results indicated that **C6** may possess high activity in growth regulation and senescence by inhibiting the signal transduction of SL.

#### 2.2.2. Parasitic Seed Germination in Response to **Cs** Application

Previous studies indicated that SL inhibitors could inhibit parasitic seed germination [7]. To test this possibility, we firstly applied target compounds at 80 μM, along with GR24 (at 100 nM) on preconditioned *P. aegyptiaca* seeds. As shown in Figure 4, all compounds could inhibit the germination of *P. aegyptiaca* seeds to varying degrees, but the compound **C6** with high tillering activity was not as prominent in inhibiting the germination of *P. aegyptiaca* seeds as in promoting rice tillering. Therefore, we speculated that the difference in shoot branching and root parasitic weed seed germination activity of **C6** may be due to the differences in the structure of the SL receptor protein, which also deserves further in-depth study in future work. Subsequently, we further compared the inhibitory effects of **C6** and DL1b on the germination of *P. aegyptiaca* seeds at different concentrations (Table 1). The results show that both **C6** and DL1b could inhibit *P. aegyptiaca* seed germination, and **C6** exhibited slightly higher activity than DL1b, with IC_50_ values of 82.8 μM and 83.5 μM, respectively.

#### 2.2.3. Bioactivity of **C6** in Dark-Induced Leaf Senescence

As mentioned above, SLs exhibit activity in accelerating leaf senescence as well. The high tillering activity of **C6** led us to investigate whether **C6** shows activity in delaying dark-induced leaf senescence in comparison to the standard SL inhibitor DL1b. DMSO was selected as the positive control here. As shown in Figure 5A, it could be observed that leaf tissues in the control group exhibited significant yellowing on the third day, while leaves in the **C6** and DL1b groups retained a large area of green. By the fourth day, rice leaves in the control group had completely yellowed and aged, while leaves in the **C6** and DL1b groups still retained varying degrees of green. On the fifth day, the green color of the leaves had largely faded to yellow. These results indicate that under treatment with a concentration of 10 μM, compounds **C6** and DL1b both delayed the aging of rice leaves, prolonging the process by one day compared to the control group. This observation was substantiated by the quantification of chlorophyll content, where the leaf segments treated with **C6** and DL1b on the third day exhibited significantly higher levels of chlorophyll compared to those in the control group (Figure 5B). These results further strengthen the notion that **C6** exhibited pronounced efficacy in delaying dark-induced leaf senescence. However, leaf senescence is not solely regulated by SL as a specific plant hormone, hence the conclusion that **C6** is an SL-related inhibitor cannot be drawn solely from this result. Combining the results of other bioactivity assays, we can only infer that **C6** may possess a corresponding function as an SL inhibitor, thereby impacting leaf senescence. Further research is needed in the future to elucidate the specific mechanism by which **C6** regulates leaf senescence.

#### 2.2.4. Effect on MDA (Malondialdehyde) Levels and SOD (Superoxide Dismutase), POD (Peroxidase), CAT (Catalase) Activities in Detached Rice Leaves in Response to **C6** Application

Considering the promising effects of **C6** in delaying leaf senescence, we conducted further evaluations to examine its impact on the antioxidant stress of leaf tissue following chemical treatment. DMSO was selected as the positive control here. As demonstrated in Figure 6A,B, as leaves progressed in age, the MDA content and POD activity in the leaf tissue gradually increased. However, the increase in MDA content observed in the leaf tissue treated with **C6** for 3 days was significantly smaller compared to the control group, indicating that the chemical treatment effectively delayed lipid peroxidation of the cell membrane. Furthermore, the CAT and SOD activity decreased as leaves progressed in age and the CAT activity in the leaf treated with **C6** and DL1b for 3 days was notably higher than that in the control group, suggesting that the chemical treatment suppressed the decline in CAT activity (Figure 6C,D). These results indicated that **C6** may contribute to the delay of leaf senescence by enhancing the protective capability against leaf membrane lipid peroxidation.

#### 2.2.5. Activity of **C6** on *Arabidopsis* Hypocotyl Elongation of Wild Type and *max2* Mutant

Several studies have shown that SLs can hinder hypocotyl elongation in *A. thaliana* [35]. In order to provide additional support for the notion that **C6** could act as a credible suppressor of SL response, we conducted supplementary experiments using both wild-type (*Col-0*) and SL-insensitive mutant (*max2-1*) plants to evaluate the effects of **C6** on hypocotyl elongation in *A. thaliana*. DMSO was selected as the positive control here. As shown in Figure 7A, **C6** and DL1b significantly increased hypocotyl elongation in wild-type seedlings at concentrations of 1–10 μM. To further verify whether **C6** was relevant to SL signal transduction, we further investigated the effects of **C6** on relieving the inhibitory effects of GR24 on hypocotyl elongation in *A. thaliana*. Furthermore, the application of **C6** and DL1b effectively counteracted the suppressive influence of GR24, highlighting the potential of **C6** as an inhibitor of SL receptors in addition to DL1b. Indeed, the application of **C6** on wild-type seedlings relieved the SL repression on hypocotyl elongation, not only in the absence but also in the presence of GR24.

In *Arabidopsis*, SLs are perceived by the α/β-hydrolase D14 that interacts with the F-box protein MAX2 and involved in the hypocotyl elongation [36,37]. To validate our hypothesis, we selected SL-insensitive mutant *max2-1* as material to further assess the effect of **C6** on hypocotyl elongation. As shown in Figure 7B, it was found that the *max2-1* mutant exhibited a longer hypocotyl phenotype compared to the wild-type, validating previous reports on this subject [38]. Furthermore, the insensitivity of *max2-1* to **C6**, in line with the effects of GR24 and DL1b, corroborated the notion that **C6** suppressed the SL signaling transduction pathway, thereby influencing hypocotyl growth in *A. thaliana.*

### 2.3. Analysis of ITC Assay

The **C6** solution was titrated with AtD14 (SL receptor in *Arabidopsis*) solution to investigate the binding capacity quantified by ITC due to the lack of the purified protein of SL receptors of OsD14 (SL receptor in rice) and ShHTL7 (SL receptor in *Sriga*). As shown in Figure 8, the thermodynamic acidity constants (Ka) of **C6** was at the same level as that of DL1b, indicating that **C6** and DL1b can both interact with the receptor protein AtD14. In addition, the negative values of ΔH and ΔS also suggest the occurrence of interaction between the small molecule and the receptor protein during titration. Therefore, we inferred that **C6** may competitively inhibit the binding of SL to the receptor protein by binding to the receptor.

### 2.4. Molecular Docking Analysis

Molecular docking assay was performed to further elucidate the mechanism of action of **C6**. Here, we docked **C6** and DL1b with the SL receptor proteins OsD14 (*O. sativa*) and ShHTL7 (*Striga*), respectively. As shown in Figure 9, **C6** and DL1b exhibited similar poses in the binding pocket of the OsD14 receptor protein, forming hydrogen bond interactions with the residue Ser220 of the receptor protein through the carbonyl group of ligand. Therefore, we proposed that **C6**, similar to DL1b, disrupts the normal binding of SL to the OsD14 receptor protein by interacting with the residue Ser220, which ultimately led to the high tillering phenotype in *O. sativa*. Based on the docking results of **C6** and DL1b with ShHTL7 (shown in Appendix A), it was found that both small molecules were not effectively surrounded within the receptor cavity, explaining their relatively poor activity in inhibiting seed germination of root parasitic weeds. Furthermore, the docking score of **C6** was −7.2 kcal/mol, which is lower than that of DL1b (−6.9 kcal/mol), indicating that **C6** exhibited a stronger binding affinity with the receptor protein ShHTL7 compared to DL1b. And this stronger binding affinity could be attributed to the hydrogen bond interaction between compound **C6** and residue CYS19. As a result, **C6** demonstrated slightly higher activity in inhibiting seed germination than DL1b.

## 3. Materials and Methods

### 3.1. Chemicals, Analytical Instruments and Plant Materials

All reagents were purchased from Innochem (Beijing, China). The solvents were dried and purified according to standard procedures and the commercially available reagents were used without further purification. Column chromatography was conducted on a silica gel plug (300–400 mesh), and the reaction progress was monitored by thin-layer chromatography on silica gel GF-254 and detected under UV light.

The plants seeds *A. thaliana* (*Col-0*), *O. sativa* (Nipponbare), SL-deficient mutant line *d10* were purchased from BIGOL CeneTech and SL-insensitive mutant line *max2-1* were provided by the State Key Lab of Plant Environmental Resilience, China Agricultural University, Beijing, China. The *P. aegyptiaca* seeds were provided by Researcher Wei He, Xinjiang Academy of Agricultural Sciences, Xinjiang, China.

The computer-aided analysis software MOE 2019 (Chemical Computing Group Inc., Montreal, QC, Canada) was supported by Professor Li Zhang, College of Science, China Agricultural University, Beijing, China. ^1^H NMR and ^13^C NMR spectra were obtained at 500 MHz using a Bruker AVANCE DPX500 spectrometer (Bruker (Beijing) Technology Co., Ltd., Karlsruhe, Germany) in CDCl_3_ solution with tetramethylsilane as the internal standard. High Resolution Mass Spectrometer (HRMS) was performed using an Agilent 6520 Accurate-Mass-Q-TOF LC/MS system (Agilent Technology Inc., Santa Clara, CA, America), equipped with an electro spray ionization (ESI) source in the positive ionization mode. The *A. thaliana* growth data were obtained using ImageJ software (https://imagej.nih.gov/ij/index.html accessed on 14 November 2022).

### 3.2. General Synthesis

The synthetic route for the target compounds is shown in Figure 1, and the characterization data of target compounds are shown in the Appendix A.

The key intermediates, 2-Bromo-1-phenylethanones, were obtained either through chemical synthesis or purchased directly [32]. 2-Bromo-1-phenylethanones were synthesized by reacting acetophenones with Br_2_ solute, and then reacting intermediates with 5-Bromo-1-naphthoic acid to afford the target compound **C1**–**C18**.

#### 3.2.1. The Preparation of 2-Bromo-1-Phenylethanones

We will take the synthesis of 2-Bromo-1-(3-chlorophenyl)ethanone as an example. In a 100 mL round bottom flask the 3′-Chloroacetophenone (1.00 g, 6.47 mmol, 1 eqv) solute in CHCl_3_ (30 mL) were slowly added. Br_2_ solute (1.03 g, 6.47 mmol, 1 eqv) in CHCl_3_ (1.00 g Br_2_/ 4 mL CHCl_3_). The reaction was stirred for 14 h at room temperature. Then a 10%-Na_2_S_2_O_3_ solution was added (10 mL) and the solution stirred for 10 min at room temperature. Then the mixture was extracted three times with ethyl acetate (3 × 10 mL). The combined organic phases were washed with brine (5 mL), dried over Na_2_SO_4_ and the solvent was removed under reduced pressure. White solid (1.24 g, 84%) was obtained and used in the next step without purification.

The other 2-Bromo-1-phenylethanones were synthesized in the same way or purchased from Innochem.

#### 3.2.2. The Preparation of Target Compounds **C1**–**C18**

2-Bromo-1-(3-chlorophenyl)ethanone (0.51 g, 2.14 mmol, 1eqv) was dissolved in dimethyl formamide (10 mL), K_2_CO_3_ (0.59 g, 4.28 mmol, 2eqv) was then added, followed by the addition of 5-Bromo-1-naphthoic acid (0.81 g, 3.21 mmol, 1.5 eqv), and stirred overnight at room temperature. The reaction mixture was quenched with H_2_O (10 mL) and extracted with EA (3 × 20 mL) and washed with H_2_O and saturated sodium bicarbonate. Then the organic layer was dried with Na_2_SO_4_, filtered and concentrated under reduced pressure. The resulting solid was purified by column chromatography. A white solid (0.67 g, 77%) was obtained as compound **C7**. m.p. 104 °C. ^1^H NMR (500 MHz, CDCl_3_) δ 8.91 (d, *J* = 8.7 Hz, 1H), 8.53 (d, *J* = 8.5 Hz, 1H), 8.37 (d, *J* = 7.2 Hz, 1H), 8.10–7.89 (m, 1H), 7.85 (d, *J* = 7.5 Hz, 2H), 7.72–7.51 (m, 2H), 7.51–7.37 (m, 2H), 5.61 (s, 2H). ^13^C NMR (126 MHz, CDCl_3_) δ 191.06, 166.52, 135.68, 135.38, 133.98, 132.76, 132.71, 132.22, 131.31, 130.68, 130.33, 128.12, 128.02, 126.94, 125.99, 125.90, 125.73, 123.32, 66.56. HRMS (ESI-Orbitrap): *m*/*z* calcd for C_19_H_13_O_3_ClBrNa [M + Na]^+^, 424.9551; found, 424.9554.

The remaining target compounds were synthesized using the same methodology, and the characterization data of ^1^H NMR, ^13^C NMR, HRMS, and melting point are shown below. The ^1^H NMR and ^13^C NMR spectra images can be found in the Appendix A.

Compound **C**1, yellow solid, yield 76%. m.p. 108 °C. ^1^H NMR (500 MHz, CDCl_3_) δ 9.05–8.79 (m, 1H), 8.63–8.46 (m, 1H), 8.47–8.30 (m, 1H), 8.12–7.97 (m, 1H), 7.96–7.76 (m, 1H), 7.72–7.49 (m, 2H), 7.51–7.38 (m, 1H), 7.36–7.26 (m, 1H), 7.23–7.11 (m, 1H), 5.56 (s, 2H). ^13^C NMR (126 MHz, CDCl_3_) δ 190.45, 166.67, 162.53 (d, *J* = 254.2 Hz), 135.84 (d, *J* = 9.1 Hz), 132.59, 132.51 (d, *J* = 62.7 Hz), 131.21, 130.95–130.92 (m), 130.66, 128.06, 127.35, 126.05, 125.88, 125.04, 125.03, 123.31, 122.45 (d, *J* = 14.5 Hz), 116.69 (d, *J* = 23.5 Hz), 69.81 (d, *J* = 14.7 Hz). HRMS (ESI-Orbitrap): *m*/*z* calcd for C_19_H_13_O_3_FBrNa [M + Na]^+^, 408.9846; found, 408.9848.

Compound **C2**, white solid, yield 60%. m.p. 140–142 °C. ^1^H NMR (500 MHz, CDCl_3_) δ 8.91 (d, *J* = 8.7 Hz, 1H), 8.52 (d, *J* = 8.5 Hz, 1H), 8.40–8.30 (m, 1H), 7.84 (d, *J* = 7.4 Hz, 1H), 7.75 (d, *J* = 7.7 Hz, 1H), 7.71–7.59 (m, 2H), 7.55–7.39 (m, 2H), 7.36–7.27 (m, 1H), 5.61 (s, 2H). ^13^C NMR (126 MHz, CDCl_3_) δ 191.05, 166.52, 162.93 (d, *J* = 248.8 Hz), 136.15 (d, *J* = 6.7 Hz), 132.73, 132.70, 132.22, 131.30, 130.74 (d, *J* = 7.8 Hz), 130.67, 128.11, 126.96, 125.99, 125.74, 123.57 (d, *J* = 4.1 Hz), 123.31, 121.13 (d, *J* = 21.5 Hz), 114.72 (d, *J* = 22.6 Hz), 66.62. HRMS (ESI-Orbitrap): *m*/*z* calcd for C_19_H_12_O_3_FBrNa [M + Na]^+^, 408.9846; found, 408.9849.

Compound **C3**, white solid, yield 62%. m.p. 144–146 °C. ^1^H NMR (500 MHz, CDCl_3_) δ 8.93 (d, *J* = 8.7 Hz, 1H), 8.54 (d, *J* = 8.5 Hz, 1H), 8.47–8.35 (m, 1H), 8.11–7.98 (m, 2H), 7.86 (d, *J* = 7.4 Hz, 1H), 7.70–7.61 (m, 1H), 7.50–7.39 (m, 1H), 7.24–7.15 (m, 2H), 5.64 (s, 2H). ^13^C NMR (126 MHz, CDCl_3_) δ 190.66, 166.68, 166.31 (d, *J* = 256.5 Hz), 132.79, 132.31, 131.35, 130.77, 130.71 (d, *J* = 4.6 Hz), 130.62, 128.16, 127.13, 125.94 (d, *J* = 29.5 Hz), 123.38, 116.32 (d, *J* = 21.8 Hz), 66.51. HRMS (ESI-Orbitrap): *m*/*z* calcd for C_19_H_12_O_3_FBrNa [M + Na]^+^, 408.9846; found, 408.9847.

Compound **C4**, white solid, yield 56%. m.p. 145–147 °C. ^1^H NMR (500 MHz, CDCl_3_) δ 8.91 (d, *J* = 8.7 Hz, 1H), 8.52 (d, *J* = 8.5 Hz, 1H), 8.36 (d, *J* = 7.3 Hz, 1H), 8.14–7.98 (m, 1H), 7.95–7.78 (m, 1H), 7.64 (t, *J* = 7.9 Hz, 1H), 7.44 (t, *J* = 8.1 Hz, 1H), 7.12–6.86 (m, 2H), 5.51 (s, 2H). ^13^C NMR (126 MHz, CDCl_3_) δ 189.01 (d, *J* = 5.5 Hz), 166.62, 166.60 (dd, *J* = 259.0, 12.4 Hz), 163.26 (dd, *J* = 256.6, 12.5 Hz), 132.98 (dd, *J* = 10.7, 4.9 Hz), 132.74, 132.66, 132.25, 131.23, 130.68, 128.09, 127.21, 126.03, 125.81, 123.33, 119.14 (dd, *J* = 14.8, 3.4 Hz), 112.98 (dd, *J* = 21.6, 3.3 Hz), 104.85 (dd, *J* = 20.0, 3.3 Hz), 69.56 (d, *J* = 14.6 Hz). HRMS (ESI-Orbitrap): *m*/*z* calcd for C_19_H_11_O_3_F_2_BrNa [M + Na]^+^, 426.9752; found, 426.9755.

Compound **C5**, white solid, yield 71%. m.p. 106 °C. ^1^H NMR (500 MHz, CDCl_3_) δ 8.82 (d, *J* = 8.7 Hz, 1H), 8.44 (d, *J* = 8.6 Hz, 1H), 8.28 (dd, *J* = 7.3, 1.5 Hz, 1H), 8.16 (s, 1H), 8.06 (d, *J* = 7.9 Hz, 1H), 7.79 (d, *J* = 7.8 Hz, 1H), 7.76 (d, *J* = 7.4 Hz, 1H), 7.62–7.52 (m, 2H), 7.40–7.28 (m, 1H), 5.57 (s, 2H). ^13^C NMR (126 MHz, CDCl_3_) δ 191.14, 166.55, 134.78, 132.87, 132.74, 132.28, 131.84 (q, *J* = 33.0 Hz), 131.36, 131.04, 130.74, 130.45 (q, *J* = 3.6 Hz), 129.76, 128.19, 126.8, 126.02, 125.73, 124.83 (q, *J* = 3.5 Hz), 123.56 (q, *J* = 272.7 Hz), 123.38, 66.62. HRMS (ESI-Orbitrap): *m*/*z* calcd for C_20_H_12_O_3_F_3_BrNa [M + Na]^+^, 458.9814; found, 458.9818.

Compound **C6**, white solid, yield 70%. m.p. 122 °C. ^1^H NMR (500 MHz, CDCl_3_) δ 8.91 (d, *J* = 8.7 Hz, 1H), 8.53 (d, *J* = 8.6 Hz, 1H), 8.37 (d, *J* = 7.3 Hz, 1H), 8.08 (d, *J* = 8.0 Hz, 2H), 7.85 (d, *J* = 7.4 Hz, 1H), 7.77 (d, *J* = 8.1 Hz, 2H), 7.64 (t, *J* = 7.9 Hz, 1H), 7.44 (t, *J* = 8.1 Hz, 1H), 5.64 (s, 2H). ^13^C NMR (126 MHz, CDCl_3_) δ 191.48, 166.54, 136.91, 135.29 (q, *J* = 32.8 Hz), 132.89, 132.74, 132.28, 131.38, 130.75, 128.30, 128.20, 126.8, 126.10 (q, *J* = 2.8 Hz), 126.03, 125.72 (q, *J* = 3.8 Hz), 123.98 (q, *J* = 192.8 Hz), 123.4, 66.69. HRMS (ESI-Orbitrap): *m*/*z* calcd for C_20_H_12_O_3_F_3_BrNa [M + H]^+^, 458.9814; found, 458.9812.

Compound **C8**, white solid, yield 56%. m.p. 136–138 °C. ^1^H NMR (500 MHz, CDCl_3_) δ 8.93 (d, *J* = 8.8 Hz, 1H), 8.61–8.51 (m, 1H), 8.43–8.36 (m, 1H), 8.01–7.91 (m, 2H), 7.89–7.81 (m, 1H), 7.71–7.62 (m, 1H), 7.56–7.49 (m, 2H), 7.49–7.43 (m, 1H), 5.64 (s, 2H). ^13^C NMR (126 MHz, CDCl_3_) δ 190.86, 166.39, 140.42, 132.58, 132.54, 132.35, 132.06, 131.12, 130.49, 129.19, 129.09, 127.93, 126.81, 125.81, 125.55, 123.14, 66.31. HRMS (ESI-Orbitrap): *m*/*z* calcd for C_19_H_12_O_3_ClBrNa [M + Na]^+^, 424.9551; found, 424.9553.

Compound **C9**, white solid, yield 58%. m.p. 160 °C. ^1^H NMR (500 MHz, CDCl_3_) δ 8.91 (d, *J* = 8.7 Hz, 1H), 8.58–8.46 (m, 1H), 8.37–8.23 (m, 1H), 7.89–7.77 (m, 1H), 7.67–7.56 (m, 1H), 7.47–7.40 (m, 1H), 7.39–7.29 (m, 3H), 5.41 (s, 2H). ^13^C NMR (126 MHz, CDCl_3_) δ 195.51, 165.86, 136.17, 132.88, 132.75, 132.21, 131.64, 131.55, 131.39, 130.68, 128.28, 128.16, 126.55, 125.94, 125.69, 123.32, 68.57. HRMS (ESI-Orbitrap): *m*/*z* calcd for C_19_H_11_O_3_Cl_2_BrNa [M + Na]^+^, 458.9161; found, 458.9166.

Compound **C10**, yellow solid, yield 70%. m.p. 136 °C. ^1^H NMR (500 MHz, CDCl_3_) δ 8.94–8.89 (m, 1H), 8.59–8.51 (m, 1H), 8.38 (dd, *J* = 7.3, 1.3 Hz, 1H), 8.11–8.05 (m, 1H), 7.89–7.79 (m, 2H), 7.69–7.58 (m, 2H), 7.50–7.41 (m, 1H), 5.60 (s, 2H). ^13^C NMR (126 MHz, CDCl_3_) δ 190.33, 166.54, 138.85, 133.92, 133.76, 132.94, 132.78, 132.32, 131.41, 131.23, 130.78, 129.99, 128.23, 126.89, 126.85, 126.05, 125.74, 123.42, 66.35. HRMS (ESI-Orbitrap): *m*/*z* calcd for C_19_H_11_O_3_Cl_2_BrNa [M + Na] +, 458.9161; found, 458.9159.

Compound **C11**, white solid, yield 79%. m.p. 120 °C. ^1^H NMR (500 MHz, CDCl_3_) δ 8.91 (d, *J* = 8.7 Hz, 1H), 8.53 (d, *J* = 8.6 Hz, 1H), 8.37 (dd, *J* = 7.3, 1.3 Hz, 1H), 8.16–8.05 (m, 1H), 7.93–7.81 (m, 2H), 7.77–7.71 (m, 1H), 7.67–7.59 (m, 1H), 7.49–7.33 (m, 2H), 5.61 (s, 2H). ^13^C NMR (126 MHz, CDCl_3_) δ 190.99, 166.51, 136.89, 135.88, 132.77, 132.70, 132.22, 131.31, 130.96, 130.68, 130.56, 128.12, 126.93, 126.34, 125.99, 125.73, 123.32, 66.53. HRMS (ESI-Orbitrap): *m*/*z* calcd for C_19_H_13_O_3_Br_2_Na [M + Na]^+^, 468.9045; found, 468.9047.

Compound **C12**, white solid, yield 70%. m.p. 120 °C. ^1^H NMR (500 MHz, CDCl_3_) δ 8.95 (d, *J* = 8.5 Hz, 1H), 8.56–8.44 (m, 1H), 8.40–8.31 (m, 1H), 8.01 (dd, *J* = 7.8, 1.8 Hz, 1H), 7.89–7.77 (m, 1H), 7.71–7.59 (m, 1H), 7.59–7.50 (m, 1H), 7.46–7.38 (m, 1H), 7.10–6.95 (m, 2H), 5.57 (s, 2H), 3.98 (s, 3H). ^13^C NMR (126 MHz, CDCl_3_) δ 192.89, 166.93, 159.63, 135.10, 132.71, 132.28, 132.20, 131.18, 131.03, 130.55, 127.91, 127.84, 126.04, 125.97, 124.32, 123.22, 121.15, 111.57, 70.69, 55.67. HRMS (ESI-Orbitrap): *m*/*z* calcd for C_20_H_16_O_4_Br [M + H]^+^, 399.0226; found, 399.0227.

Compound **C13**, white solid, yield 66%. m.p. 102 °C. ^1^H NMR (500 MHz, CDCl_3_) δ 8.92 (d, *J* = 8.7 Hz, 1H), 8.49 (d, *J* = 8.5 Hz, 1H), 8.40–8.31 (m, 1H), 7.82 (d, *J* = 7.4 Hz, 1H), 7.66–7.57 (m, 1H), 7.54–7.47 (m, 2H), 7.46–7.33 (m, 2H), 7.18–7.08 (m, 1H), 5.63 (s, 2H), 3.82 (s, 3H). ^13^C NMR (126 MHz, CDCl_3_) δ 191.96, 166.64, 160.03, 135.45, 132.69, 132.57, 132.18, 131.24, 130.63, 129.96, 128.05, 127.23, 126.01, 125.83, 123.28, 120.54, 120.26, 112.12, 66.78, 55.51. HRMS (ESI-Orbitrap): *m*/*z* calcd for C_20_H_16_O_4_Br [M + H]^+^, 399.0226; found, 399.0227.

Compound **C14**, white solid, yield 64%. m.p. 132 °C. ^1^H NMR (500 MHz, CDCl_3_) δ 8.94 (d, *J* = 8.7 Hz, 1H), 8.50 (d, *J* = 8.5 Hz, 1H), 8.37 (dd, *J* = 7.3, 1.3 Hz, 1H), 8.03–7.88 (m, 2H), 7.83 (d, *J* = 7.4 Hz, 1H), 7.67–7.57 (m, 1H), 7.52–7.31 (m, 1H), 7.04–6.87 (m, 2H), 5.61 (s, 2H), 3.86 (s, 3H). ^13^C NMR (126 MHz, CDCl_3_) δ 190.52, 166.73, 164.15, 132.70, 132.51, 132.19, 131.23, 130.61, 130.15, 128.02, 127.37, 127.20, 126.02, 125.88, 123.26, 114.15, 66.45, 55.57. HRMS (ESI-Orbitrap): *m*/*z* calcd for C_20_H_16_O_4_Br [M + H]^+^, 399.0226; found, 399.0226.

Compound **C15**, white solid, yield 61%. m.p. 124 °C. ^1^H NMR (500 MHz, CDCl_3_) δ 8.95 (d, *J* = 8.7 Hz, 1H), 8.50 (d, *J* = 8.5 Hz, 1H), 8.36 (d, *J* = 7.2 Hz, 1H), 7.84 (d, *J* = 7.4 Hz, 1H), 7.63 (t, *J* = 7.9 Hz, 1H), 7.56–7.46 (m, 1H), 7.49–7.37 (m, 1H), 7.16–7.03 (m, 1H), 6.96–6.79 (m, 1H), 5.57 (s, 2H), 3.93 (s, 3H), 3.80 (s, 3H). ^13^C NMR (126 MHz, CDCl_3_) δ 192.56, 166.96, 154.28, 153.77, 132.70, 132.26, 132.19, 131.00, 130.55, 127.90, 126.04, 125.96, 124.32, 123.22, 122.42, 113.60, 113.08, 70.76, 56.08, 55.85. HRMS (ESI-Orbitrap): *m*/*z* calcd for C_21_H_18_O_5_Br [M + H]^+^, 429.0332; found, 429.0333.

Compound **C16**, white solid, yield 59%. m.p. 142-144 °C. ^1^H NMR (500 MHz, CDCl_3_) δ 8.98–8.92 (m, 1H), 8.58–8.50 (m, 1H), 8.39 (dd, *J* = 7.3, 1.2 Hz, 1H), 7.93–7.88 (m, 2H), 7.88–7.83 (m, 1H), 7.69–7.63 (m, 1H), 7.52–7.42 (m, 1H), 7.36–7.30 (m, 2H), 5.66 (s, 2H), 2.45 (s, 3H). ^13^C NMR (126 MHz, CDCl_3_) δ 191.72, 166.75, 145.08, 132.77, 132.61, 132.27, 131.80, 131.28, 130.67, 129.69, 128.07, 128.01, 127.37, 126.06, 125.91, 123.31, 66.68, 21.87. HRMS (ESI-Orbitrap): *m*/*z* calcd for C_20_H_16_O_3_Br [M + H]^+^, 383.0277; found, 383.0274.

Compound **C17**, white solid, yield 79%. m.p. 96 °C. ^1^H NMR (500 MHz, CDCl_3_) δ 8.90 (d, *J* = 8.7 Hz, 1H), 8.51 (d, *J* = 8.5 Hz, 1H), 8.40–8.26 (m, 1H), 7.84 (d, *J* = 7.4 Hz, 1H), 7.66–7.57 (m, 1H), 7.48–7.38 (m, 1H), 5.15 (s, 2H), 2.13–1.94 (m, 1H), 1.25–1.12 (m, 2H), 1.07–0.97 (m, 2H). ^13^C NMR (126 MHz, CDCl_3_) δ 203.55, 166.43, 132.69, 132.63, 132.21, 131.14, 130.64, 128.06, 127.09, 125.95, 125.75, 123.28, 69.01, 17.42, 11.50. HRMS (ESI-Orbitrap): *m*/*z* calcd for C_16_H_14_O_3_Br [M + H]^+^, 333.0121; found, 333.0122.

Compound **C18**, white solid, yield 51%. m.p. 80 °C. ^1^H NMR (500 MHz, CDCl_3_) δ 8.88 (d, *J* = 8.7 Hz, 1H), 8.49 (d, *J* = 8.5 Hz, 1H), 8.30 (d, *J* = 7.2 Hz, 1H), 7.83 (d, *J* = 7.4 Hz, 1H), 7.61 (t, *J* = 7.9 Hz, 1H), 7.43 (t, *J* = 8.0 Hz, 1H), 5.20 (s, 2H), 1.28 (s, 9H). ^13^C NMR (126 MHz, CDCl_3_) δ 207.78, 166.68, 132.63, 132.43, 132.18, 131.07, 130.59, 127.97, 127.43, 125.98, 125.81, 123.23, 65.18, 43.00, 26.30. HRMS (ESI-Orbitrap): *m*/*z* calcd for C_17_H_18_O_3_Br [M + H]^+^, 349.0434; found, 349.0438.

### 3.3. Biological Assay

#### 3.3.1. *O. sativa* Branching Assays

*O. sativa* seedlings were cultivated hydroponically following the established methodology [39]. The seeds were subjected to sterilization in a 1% NaClO solution supplemented with 0.1% Tween 20 for 20 min. Subsequently, the seeds underwent a series of eight washes with sterilized water and were then incubated in a container filled with water at 30 °C in the dark for 48 h. The sprouted seeds were transplanted into a hydroponic nutrient solution (pH carefully adjusted to 5.5–5.8 with phosphate buffer solution) and were allowed to grow under a 16-h light and 8-h dark cycle at 28 °C for 7 days. Equally grown seedlings were subjected to different treatments with or without the targeted compounds while being maintained under identical conditions. Following a 4-week incubation period (in which the nutrient solution was renewed twice a week), the number of tillers measuring at least 5 mm in length was quantified.

#### 3.3.2. Seed Germination Bioassays against *P. aegyptiaca*

The germination bioassay for parasitic weeds was performed in accordance with previously described methodologies [40]. The seeds of the parasitic weeds *P. aegyptiaca* were subjected to surface sterilization with 75% ethanol (*v*/*v*) for 2 min, followed by thorough rinsing with sterile water and subsequent air-drying. Glass fiber filter paper discs (10 mm, Grade GF/D, Whatman, GE Healthcare, Buckinghamshire, UK) were placed in Petri dishes containing a filter paper ring immersed in sterile water, which served as a substrate for the pretreated seeds (40–60 seeds per disc). The Petri dishes were sealed and pre-incubated in darkness at 25 °C for 7 days to facilitate preconditioning prior to experimentation. After preconditioning, the discs were pretreated with varying concentrations (0, 40, 80, 160, 320 μM) of the target compound **C6** and DL1b, supplemented with 100 nM GR24. Subsequently, the Petri dishes were sealed and kept in the dark under identical conditions for 7 days. The number of germinated seeds was counted under a binocular microscope, and the germination rate (%) was calculated. The IC_50_ values (half-maximal concentration) of chemicals were calculated using the SPSS 20 software (Statistical Product and Service Solutions, Chicago, IL, USA). All treatments were replicated three times.

#### 3.3.3. Analysis of *A. thaliana* Hypocotyl Elongation

Seeds of *A. thaliana* were subjected to surface sterilization with a 2.5% NaClO solution for 15 min, followed by thorough washing with sterile water 6–8 times. The pretreated seeds were kept in darkness at 4 °C for a period of 3 days prior to utilization. Subsequently, the treated seeds were transferred onto 1/2 Murashige and Skoog (MS) culture medium supplemented with the desired concentrations of chemicals, including 1% (*w*/*v*) sucrose, 0.8% (*w*/*v*) agar and 0.1% DMSO, adjusting pH to 5.8–6.0. The cultures were subjected to continuous low light at a temperature of 22 °C for 7 days. The length of hypocotyl was measured using the ImageJ software (https://imagej.nih.gov/ij/index.html accessed on 14 November 2022) after completion of the incubation period.

#### 3.3.4. Dark-Induced Rice Leaf Senescence

Rice seeds (*Nipponbare*) were surface sterilized with 2.5% sodium hypochlorite solution and 0.05% Tween-20, and germinated on moist filter paper in the sealed petri plates. The Petri plates with germinated seeds were transferred to white fluorescent light (130–180 mM m^−2^ s^−1^) with 16 h:8 h (L/D) at 28 °C, to establish seedlings for 1 week. Seven-day-old uniform seedlings were selected and transferred to 50 mL tubes containing half strength modified Hoaglands nutrient solution. After 1 week, 2 cm leaf segments were cut from the middle part of the third leaves of rice plants. Each segment was put in a well (in 12-well plates) containing 4 mL of 3.0 mM MES buffer with 0.05% Tween-20. **C6** and DL1b were applied at 10 μM concentration. Plates were incubated at 28°C in the dark for 5 days. After application of **C6** and DL1b, changes in leaf color were monitored on a daily basis, and chlorophyll content on the 3rd and 5th day were monitored.

Determination of chlorophyll content: 0.5 g of the penultimate fully expanded leaf was cut into fine strips. The samples were extracted with 50 mL of 95% ethanol under light-shielded conditions for 72 h. The absorbance of the sample at wavelengths 470 nm, 649 nm, and 665 nm were then measured using a UV spectrophotometer.

The following formulas are used for calculation:Ca = (13.95 ∗ D665 − 6.88 ∗ D649) ∗ V/(1000 ∗ W), Cb = (24.96 ∗ D649 − 7.32 ∗ D665) ∗ V/(1000 ∗ W), Ce = [(1000 ∗ D470 − 2.05 ∗ Ca − 114.8 ∗ Cb)/245] ∗ V/(1000 ∗ W).

Ca, Cb, and Ce represent the concentrations of chlorophyll a, chlorophyll b, and other chlorophylls, respectively. D649, D665, and D470 represent the absorbance values of the extraction solution at 649 nm, 665 nm, and 470 nm, respectively. V represents the volume of the extraction solution (mL). W represents the fresh weight of the sample (g).

It is important to ensure all procedures are conducted under light-shielded conditions to avoid any potential effects of light exposure on the measurements.

#### 3.3.5. Determination of SOD, POD, CAT, and MDA Levels in Detached Rice Leaves

One-week-old rice seedlings were established as mentioned above. Uniform seedlings were transferred to 50 mL tubes containing half-strength Hoagland nutrient solution for 7 d and grown in an incubator under white fluorescent light with 16 h/8 h (L/D) at 28 °C for 2 months. Leaf segments of 2 cm were cut from the middle part of the second-to-last leaves of rice plants. Each segment was put in a well (in 24-well plates) containing 2 mL of 2.5 mM MES buffer with 0.05% Tween-20, and incubated at 30 °C in the dark. After application of **C6** and DL1b for 3 days and 5 days, SOD, POD, CAT, and MDA levels were counted.

The content of malondialdehyde (MDA) was measured using the thiobarbituric acid (TBA) method, while the activities of catalase (CAT) and peroxidase (POD) were determined using visible spectrophotometry and colorimetric methods, respectively, using reagent kits provided by Nanjing Jiancheng Institute of Biotechnology. The activity of superoxide dismutase (SOD) was measured using a kit provided by Box Bio.

### 3.4. Isothermal Titration Calorimetry (ITC) Assay

The purified AtD14 protein was buffer-exchanged using a centrifugal filter unit with a buffer composed of 50 mM Tris-HCl (1 M, pH 7.5) and 150 mM NaCl to remove imidazole. The protein concentration was adjusted to 1 μM. In the sample cell, 300 μL of protein solution was added, and the titrant needle was loaded with 250 μM compound, a total of 19 injections were performed. The initial injection volume was 0.4 μL, the injection duration was 2 s, and the subsequent injection volume was 2 μL. The duration of each injection was 4 s with a 180 s interval between each pair of injections. The data output frequency was set to 5 s, the sample cell temperature was 25 °C, the reference power was 10 μCal/s and there was a 60 s interval from the beginning of the experiment to the first injection and the stirrer speed was set to 1000 RPM. Detailed methods for constructing and purifying the AtD14 protein are given in the Appendix A.

### 3.5. Molecular Docking

Molecules were drawn with ChemBioDraw Ultra 13.0 software and minimized with MOE 2019 (Chemical Computing Group Inc., Montreal, QC, Canada). The SL receptor OsD14 (PDB: 5DJ5) and ShHTL7 (PDB: 7SNU) downloaded from Protein Data Bank (https://www.rcsborg accessed on 2 April 2023) were prepared by the process of deleting water, adding hydrogen, adding Gasteiger charges, merging nonpolar hydrogen and so on by AutodockTools-1.5.6 (Scripps, La Jolla, CA, USA). Docking was operated by MOE 2019 after setting method (placement: triangle matcher, refinement: rigid receptor), score (placement: London dG, refinement: GBVI/WSA dG) and poses (placement: 300, refinement: 5). The lowest binding energy for the docked conformations was chosen from 300 conformations as the representative binding energy to evaluate the potential of the corresponding compounds. The best docking poses were selected for analyzing the interactions between SL receptor and target compounds.

## 4. Conclusions

In conclusion, following the design and preparation of 18 novel phenyl-substituted naphthoic acid ethyl ester derivatives, their promotive activity on rice tillering was tested, and **C6** was found to be more active than DL1b. Further investigations were conducted on *P. aegyptiaca* seed germination, rice leaf senescence, and *A. thaliana* hypocotyl elongation activities of **C6**, which revealed its biologically active nature as an SL inhibitor. The mechanism of action of **C6** was further explored through ITC and molecular docking experiments, and the results demonstrated that **C6** can indeed interact with the SL receptor protein and bind to the SL receptor D14 in a similar pose, which may help in understanding the different functions of SLs and have a large application potential in agriculture, particularly in the regulation of plant architecture.

## Data Availability

The datasets used and/or analyzed during the current study are available from the corresponding author on reasonable request.

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
