# Peer review of "Design, Synthesis and Biological Evaluation of Novel Phenyl-Substituted Naphthoic Acid Ethyl Ester Derivatives as Strigolactone Receptor Inhibitor"

_ijms, 2024, doi:10.3390/ijms25073902_

Round 1
Reviewer 1 Report
Comments and Suggestions for Authors
Dear Editor and the Authors,
The paper reports the design, synthesis and characterization of DL1b analogs targeting strigolactone receptor, which was tested in the study. The developed molecules were applied to a crop to analyze how they will affect the yield. Besides, biochemical studies were performed in order to understand whether there was a correlation between the phenotypical improvements and biochemical improvements. Also, the paper reports the capacity of the developed molecule against parasite. For sure, the findings and chemistries are new and worth to publish. There should be only minor improvements in the paper;
1- In abstract Phelipanche aegyptiaca should be written in italic.
2- "Thermodynamic acid constants" should be replaced with "Thermodynamic acidity constants"
3- Supplementary file was not reachable to me, so I couldn't see the NMR spectra. Besides, the NMR spectra should be provided and shifts relating to each group should be shown on the molecules inserted into the spectra. This can help researchers working on the similar topics. This can also increase the attractiveness of the paper.
4- There should be a table comparing the powerful C6 molecule with the published successful DL1b analogous.
Kind Regards,
Author Response
- In abstract Phelipanche aegyptiaca should be written in italic.
Response:
Thank you for your guidance regarding our manuscript. We have now corrected the error in the abstract (line 21-22).
- "Thermodynamic acid constants" should be replaced with "Thermodynamic acidi
Response:
We sincerely thank reviewer raising this point.We have corrected the description in the manuscript (lin 235).
- Supplementary file was not reachable to me, so I couldn't see the NMR spectra. Besides, the NMR spectra should be provided and shifts relating to each group should be shown on the molecules inserted into the spectra. This can help researchers working on the similar topics. This can also increase the attractiveness of the paper.
Response:
We greatly appreciate the suggestions you have made. In the latest manuscript, we will be re-uploading the SI. However, due to the combined influence of factors such as experimental conditions, sample properties, and instrument characteristics, achieving precise correlation of each peak’s chemical shift with its corresponding atom is not always straightforward. Considering the inherent uncertainties in calibration and the potential for errors, we have chosen not to explicitly label the chemical shifts and their corresponding atoms on the spectrum.
- There should be a table comparing the powerful C6 molecule with the published successful DL1b analogous.
Response:
We greatly appreciate the suggestions you have made. In the latest draft, we have depicted the molecular structures of C6 and DL1b in Figure 3.
Reviewer 2 Report
Comments and Suggestions for Authors
The paper is interesting and well written. The research is original, well designed and well performed and the results are relevant.
I only have some minor points:
Line 21: binnomial name should be in italics.
Line 78: Molecular without capitals.
Figure 3: Enlarge. Difficult to read.
Line 119: germination.
Fig 4: enlarge. Lettering too small. The nambers above each bar are quite obvious, I recommend to delete them.
Table 1: Please include an statistical analysis to check the significacnce of the differences.
Fig 6: The legend is confussing and contains some mistakes. Should be written as: "Effect on MDA levels and POD, CAT and SOD activities".
Fig 7: Enlarge.
Author Response
Reviewer 2
The paper is interesting and well written. The research is original, well designed and well performed and the results are relevant.
I only have some minor points:
Line 21: binnomial name should be in italics.
Response:Thank you for your guidance regarding our manuscript. We have now corrected the error in the abstract (line 21-22).
Line 78: Molecular without capitals.
Response:Thank you reviewer for pointing this out. We have now corrected the error in the abstract (line 82).
Figure 3: Enlarge. Difficult to read.
Response:Thans for the suggestion. We have redrawn Figure 3, adding the chemical structures of compounds C6 and DL1b. Additionally, we have adjusted the clarity of the image for better visualization.
Line 119: germination.
Response:We feel sorry for such error and we have corrected the spelling of germination(Line 128)
Fig 4: enlarge. Lettering too small. The nambers above each bar are quite obvious, I recommend to delete them.
Response:Thank you for your advice. We have redrawn the Fig 4 and deletded the numbers above each bar.
Table 1: Please include an statistical analysis to check the significacnce of the differences.
Response:Thank you very much for pointing this out, the difference in germination rates between the two compounds has been reanalyzed.
Fig 6: The legend is confussing and contains some mistakes. Should be written as: "Effect on MDA levels and POD, CAT and SOD activities".
Response:Thank you for pointing that out, we have made corrections in the respective section.(line 181-183,line 198)
Fig 7: Enlarge.
Response:Thank you for pointing that out and we have redrawn Figure 7 and adjusted the title and annotation
Reviewer 3 Report
Comments and Suggestions for Authors
The work is interesting and fits in with new tools for agriculture.
However brevity in many cases made it difficult to understand for this reviewer. The main issue was in the legends for figures - information was inadequate. Quality of images needs to be improved
Much of the writing would be OK if each reader worked daily with SLs but this is not the case for this reviewer.

Comments on the Quality of English Languagefew issues not many well done here
Author Response
Reviewer 3
The work is interesting and fits in with new tools for agriculture.
However brevity in many cases made it difficult to understand for this reviewer. The main issue was in the legends for figures -information was inadequate. Quality of images needs to be improved.
Much of the writing would be OK if each reader worked daily with SLs, but this is not the case for this reviewer.
Response:First of all, we sincerely thank you for the suggestions you provided for this manuscript, which greatly contribute to enhancing and improving the quality of our manuscript. In response to the suggestions you offered, we have made the following modifications, which have been highlighted in the latest MS.
- We have checked the Abstract and corrected the errors in the corresponding part.(line 22-23)
- We have made the corresponding modifications to explain the synthesis and distribution sites of SL in plant organisms.(Line 33-34: SLs are primarily synthesized in the roots of plants, but there is also a small amount distributed in stems and leaves.)
- We have redrawn Figure 1 and added the chemical structures of some natural SLs to Figure 1.
- In Figure 2, we have supplemented relevant legends, and we have rephrased the title of Figure 2as follow: Figure 2. Design scheme of target compounds (upper: chemical structures of some highly active drugs containing benzene rings, lower: design of target compounds C1-C18 from DL1b).
- At this point an image showing a natural inhibitor and these compounds would be useful
Response: Thank you for you suggestion. It’s worth noting that in current literature, there hasn’t been mention of naturally SL inhibitors. Both the compounds mentioned in existing literature and those discussed in this paper as SL inhibitors are artificially synthesized
- We have provided a revised introduction to the relevance of the CCD8 gene in the biosynthesis of SLs to aid readers in understanding our manuscript(Line 100-101:As previously reported, Carotenoid cleavage dioxygenase CCD8 plays a crucial regulatory role in the biosynthetic pathway of SLs in plants [33, 34]).
- About Figure 3:We have redrawn Figure 3, adding the chemical structures of compounds C6 and DL1b. Additionally, we have adjusted the clarity of the image for better visualization.
- About Figure 4: In the absence of germination-inducing strigolactones (SLs), the natural germination rate of aegyptiacaseeds is extremely low. Additionally, due to the significant technical challenges associated with extracting natural SLs from root exudates of the species, we employed the highly active SL analog GR24 to mimic the stimulatory effect of plant-secreted SLs on P. aegyptiaca seeds. Consequently, when the signaling transduction of SLs is disrupted, the germination of P. aegyptiaca seeds is inhibited due to their inability to perceive SLs.
In this study, we evaluated the association of compound C6 with the signaling transduction pathway of strigolactones by comparing the germination response of P. aegyptiaca seeds treated with C6 and DL1b under the induction of GR24.
- Additional notes have been added to Table 1 as follows: Germinated and non-germinated seeds were counted from each disc to calculate percentagegermination.Data are means ± SE (n = 5 discs). IC50: Half-maximal inhibitory concentration.
- Given that C6 has been demonstrated to possess activities in promoting tillering and inhibiting the germination of root parasitic seeds, we would like to further investigated whether C6 also plays a regulatory role in leaf senescence similar to SLs. Here, we have made modifications and additions to Section 2.2.3 to assist readers in better understanding our manuscript.(Line 161, Line 167-169)
- In section 2.2.4, Firstly, the SOD and POD activities we measured refer to total enzyme activity. Additionally, our aim was to preliminarily investigate whether C6 and DL1b would affect the senescence process of leaves. Till the fifth day, the leaves had already turned completely yellow, so we only recorded the senescence status of the leaves from 0 to 5 days. Regarding the question raised by the reviewer about whether leaves placed under light after turning completely yellow would undergo changes, we did not delve deeply into this aspect. However, this also provides us with a valuable direction for further exploration. In subsequent in-depth studies, we will attempt to explore this direction and integrate other tools such as transcriptomic analysis to elucidate the specific mechanisms by which C6 and DL1b influence leaf senescence.
- About 2.2.5 section: Firstly, we have redrawn Figure 7 and corrected the title and annotation. Secondly, we have provided brief supplementary information on the gene Max2, introducing its association with SL signaling to explain why we used max2mutants as materials for related validation experiments. Specific modifications have been highlighted in the manuscript.(line 222-225)
- In 2.3 section, we have annotated the relevant information on AtD14 to assist readers in better understanding our manuscript.(line 232-234)
Round 2
Reviewer 3 Report
Comments and Suggestions for Authors
thanks for modifications
but there are still many comments that could be addressed Please look at all sticky notes
many small grammatical problems
major problem is with assays on excised materials where the excision is not made apparent in text and would cause changes in JA, ethylene SA etc
ie the studies are with plant material that is not native to an intact plant but is responding to intense wounding before and as the SLs and inbitors are applied this should be discussed

Comments on the Quality of English Languageneeds work editing for scientific acceptability and also clarity of wording
Author Response
Dear reviewer: Thank you very much for your suggestions. We have responded to the comments in the PDF file ijms-2915918-review-2, and the specific changes have been presented and marked in the MS.
In the latest MS, we have rephrased the methods for culturing and obtaining excised leaf materials, as well as adjusted and rephrased some sentences in the manuscript, with specific details annotated in the text. Additionally, our intent in evaluating the activity of C13 in regulating leaf senescence was to explore whether C13 might have a biological function as an SL inhibitor, enriching our research content. As for the specific mechanism by which C13 regulates leaf senescence, it is also worth further exploration and will be extensively investigated in our future work.
Thank you for you useful advice again.
